# Effects of Deep Brain Stimulation on Autonomic Function

**DOI:** 10.3390/brainsci6030033

**Published:** 2016-08-16

**Authors:** Adam Basiago, Devin K. Binder

**Affiliations:** 1School of Medicine, University of California, Riverside, CA 92521, USA; adam.basiago@email.ucr.edu; 2Division of Biomedical Sciences, School of Medicine, University of California, 1247 Webber Hall, Riverside, CA 92521, USA

**Keywords:** deep brain stimulation, autonomic dysfunction, subthalamic nucleus, periaqueductal or periventricular gray, globus pallidus interna, thalamus, blood pressure, sweating, micturition, gastrointestinal motility

## Abstract

Over the course of the development of deep brain stimulation (DBS) into a well-established therapy for Parkinson’s disease, essential tremor, and dystonia, its utility as a potential treatment for autonomic dysfunction has emerged. Dysfunction of autonomic processes is common in neurological diseases. Depending on the specific target in the brain, DBS has been shown to raise or lower blood pressure, normalize the baroreflex, to alter the caliber of bronchioles, and eliminate hyperhidrosis, all through modulation of the sympathetic nervous system. It has also been shown to improve cortical control of the bladder, directly induce or inhibit the micturition reflex, and to improve deglutition and gastric emptying. In this review, we will attempt to summarize the relevant available studies describing these effects of DBS on autonomic function, which vary greatly in character and magnitude with respect to stimulation target.

## 1. Introduction

Deep brain stimulation (DBS) has evolved into a well-established therapy for Parkinson’s disease [1], essential tremor [2], and dystonia [2], as well as a therapy for multiple sclerosis [3], cluster headache [4], Tourette syndrome [5], and obsessive-compulsive disorder [6]. DBS is even being investigated as a surgical intervention for obesity, major depression, and a therapy for restoring memory to patients with Alzheimer disease [7,8,9,10,11]. In addition to the primary symptoms treated by DBS, many groups have investigated its effect on autonomic functions at various target sites in the brain [12,13]. Dysfunction of autonomic processes is common in neurological diseases [14,15,16,17,18]. In Parkinson’s disease and multiple sclerosis, for example, patients are afflicted with varying manifestations of dysautonomia including orthostatic and cardiovascular dysregulation, lower urinary tract dysfunction, sudomotor dysfunction, and gastrointestinal disturbances [15,17,19,20,21]. In this review, we will attempt to summarize the research available describing the effects of DBS on autonomic function.

## 2. Methodology

A PubMed search of the available literature describing the autonomic effects of DBS was conducted through EndNote using the keywords listed above; along with any applicable iterations; in order to capture as many relevant references as possible. A total of 99 references were found with this method. Those references were then categorized by the affected autonomic function and then further by DBS target. 75 of 99 references were used in the final reference list.

## 3. Sympathetic Autonomic Modulation

### 3.1. DBS and Cardiorespiratory Control

DBS has shown significant effects on hypertension and hypotension. Depending on the target in the brain, DBS can cause a decrease or increase in blood pressure, and a decrease in orthostatic hypotension [22,23,24]. What makes this variable outcome possible is precise placement of the electrodes into their targets. The region of the brain that has shown the most promise as a target for DBS blood pressure regulation is the periventricular/periaqueductal gray matter (PVG/PAG) of the midbrain, as described by Green et al. [25] in 2005. In a study of 15 patients undergoing PVG/PAG DBS for uncontrolled neuropathic pain, they observed variable changes in BP depending on whether PVG/PAG stimulation was dorsal or ventral. In six patients with ventral PVG/PAG stimulation, a mean reduction of systolic BP of 14.2 ± 3.6 mmHg (13.9%), a mean reduction of diastolic BP of 4.9 ± 2.9 mmHg (6%), and a mean reduction of pulse pressure of 9.3 ± 3.16 mmHg were observed [25]. In seven patients with dorsal PVG/PAG stimulation, a mean increase in systolic BP of 16.73 ± 5.9 mmHg (16.4%), a mean increase in diastolic BP of 4.9 ± 2.8 mmHg (6%), and a mean increase in pulse pressure of 11.83 ± 5.4 mmHg were observed [25]. There were patients in whom the electrodes did not produce any BP alteration, but it was determined that electrode location was not in the PVG/PAG. Green et al. [25] determined these effects to be due to modulation of sympathetic activity, due to the presence of changes in both total peripheral resistance (TPR) and myocardial contractility [25]. Later publications, reviews, and case reports by Green et al. and other groups all report similar findings in humans and in animal models [12,13,22,23,26,27,28,29,30,31,32].

Posterior hypothalamic area (PHA) DBS, PVG/PAG DBS and subthalamic nucleus (STN) DBS have all been shown to affect orthostatic hypotension (OH) and baroreflex sensitivity (BRS), most likely through changes in sympathetic activity. During head-up tilt testing (HUTT), PHA DBS can increase diastolic BP and TPR without changing the effect of the baroreflex on other cardiovascular parameters or resting supine BP and heart rate (HR) [33]. In 2006, Green et al. [24] showed that through an increase in BRS, PVG/PAG DBS can prevent the drop in BP on standing in patients diagnosed with OH and mild orthostatic intolerance, but does not cause resting hypertension in those patients or hypertension in the control group with no postural BP problems. A later 2014 publication by Sverisdóttir et al. [34] reported differential changes in patients with dorsolateral versus ventrolateral PAG DBS; that same publication also showed an increase in orthostatic tolerance in Parkinson’s disease (PD) patients with STN DBS. Neither BRS nor BP were influenced with stimulation of the motor thalamus, globus pallidus interna (GPi), pedunculopontine nucleus (PPN), sensory thalamus, or anterior cingulate cortex (ACC) [34]. This seems to confirm prior research by Stemper et al. [35] that showed that with stimulation during HUTT, PD patients with STN DBS had stable BP and BRS, yet without simulation during HUTT, the same patients experienced significant orthostatic hypotension. Therefore, the available data indicate some anatomic specificity to the effects of DBS on BP and BRS.

In spite of evidence supporting the sympathetically-mediated improvement of OH caused by STN DBS, its direct effects on the cardiovascular system remain unclear. A recent report of STN DBS in PD patients by Furgala et al. [36] found that STN DBS results in activation of the sympathetic nervous system resulting in changes to BP and heart rate variability. However, Trachani et al. [37] reported the opposite in 2012. Several other publications also offer conflicting opinions on the cardiovascular effects of STN DBS [38,39,40]. Sumi et al. [41] imply that the cardiovascular improvements seen with STN DBS are due not to the stimulation itself, but rather to an increased ability to exercise, thus improving overall cardiovascular health and lower extremity muscle strength. One explanation for the conflicting reports may be that the autonomic effects are not a direct result of the stimulation at all, but are rather the result of reduced need for pharmacotherapy to combat motor symptoms of PD [42]. Hyam et al. [12] suggest that because STN DBS generally requires higher frequencies and higher total energy delivery than DBS of other targets, a spread of stimulation to nearby components of the central autonomic network could be the cause rather than stimulation of the STN itself. This conclusion seems to be supported by the earlier findings of Lipp et al. [43] in 2005. In a study of five patients undergoing bilateral STN DBS for Parkinson’s disease, four patients with magnetic resonance imaging (MRI)-confirmed correct placement of their electrodes within the STN experienced no autonomic symptoms. However, in the fifth patient whose electrodes were shown by MRI to extend into the posteromedial and lateral hypothalamic areas, significant autonomic changes were observed including changes to blood pressure regulation, sweating, and breathing pattern [44]. This report emphasizes the importance of precise placement of the electrodes and the tuning of stimulation frequency and total energy delivery when autonomic effects are desired or not desired, regardless of other clinical goals.

For DBS modulation of autonomic respiratory control, the evidence is relatively new. In 2012, Hyam et al. [45] studied the effects on two pulmonary function tests—peak expiratory flow rate (PEFR) and forced expiratory volume in one second (FEV_1_)—of PAG DBS in ten neuropathic pain (NP) patients, sensory thalamus DBS in seven NP patients, STN DBS in 10 movement disorder patients, and GPi DBS in 10 movement disorder patients. Using sensory thalamus DBS to control for the effect of pain relief, and GPi DBS to control for improvement in general motor function (both with no change in PEFR), they showed that PAG DBS and STN DBS both increase PEFR [45]. There was no change in FEV_1_ with any of the stimulated targets, indicating that the increase in PEFR was likely due to bronchodilation of the large airways [45]. Further research into these effects could lead to DBS-mediated treatment of both asthma and obstructive sleep apnea through dilation of bronchioles and maintenance of upper airway patency, respectively [9]. DBS has also been implicated in increasing the respiratory rate in human and animal studies through stimulation of the anterior limb of the internal capsule and the caudal dorsal PAG [9,46,47]. A report by Vigneri et al. [48] claimed to show that DBS of the STN or PHA does not affect respiratory rate, HR or BP, but they were not able to precisely localize the electrode placement, a significant confounding factor as shown by numerous reports described above.

### 3.2. DBS and Sudomotor Control

Sudomotor dysfunction, most often hyperhidrosis, is extremely common in PD patients and has been shown to be alleviated by STN DBS [43,49,50,51]. In 2007, Witjas et al. [43] conducted a study of 30 male and 10 female patients with PD lasting an average of 12.4 ± 4.5 years, in which their nonmotor symptoms (NMS) were analyzed before and after bilateral STN DBS. One year after surgery, 34 of 35 patients were completely relieved of the drenching sweats they had experienced prior to STN DBS [43]. This effect was again seen in a later case report by Sanghera et al. [50], in which a STN DBS patient would experience whole body drenching sweats that would be alleviated with stimulation, and would return when stimulation was turned off. A study of nineteen STN DBS patients by Trachani et al. [49] observed a post-implantation reduction in hyperhidrosis in four patients, as well as an improvement of hypohidrosis in two patients. In a 2011 study of PD patients with STN DBS, Halim et al. [52] observed complete resolution of sudomotor dysfunction (and other autonomic dysfunction to be discussed later) in the three patients with early onset PD (EOPD), whereas the other eight patients with late onset PD (LOPD) did not experience any improvement in their dysautonomia. One of the three EOPD patients experienced bilateral resolution of his excessive sweating even though he only had unilateral left STN DBS [51]. Although STN DBS appears to dramatically help sudomotor dysfunction, DBS of other targets can also make it worse. A DBS electrode mistakenly placed in the thalamus or posterolateral hypothalamus can actually cause hyperhidrosis in patients who did not suffer from it prior to surgery [44,53].

## 4. Parasympathetic Autonomic Control

### 4.1. DBS and Micturition

Lower urinary tract symptoms (LUTS) are extremely common in neurological diseases like PD [54,55,56] and multiple sclerosis (MS) [3], and are a significant source of morbidity [52]. In parallel with the other dysautonomias above, DBS can either induce or inhibit micturition, dependent upon the brain target. Basal ganglia and brainstem targets (STN and PAG) appear to inhibit micturition and improve urinary incontinence, while thalamic targets (ventral intermediate and ventral posterolateral nuclei—VIM and VPL, respectively) induce micturition [3]. The vast majority of studies on DBS and micturition are on STN DBS. In 2003, Finazzi-Argò et al. [57] studied urodynamics in 5 patients with PD and LUTS following STN DBS and found that all patients experienced increased volumes for initial desire to void (V_ID_) and bladder capacity (V_BC_), as well as decreased hyperreflexive detrusor contraction. In their study of sixteen STN DBS patients with PD but no preexisting urinary problems, Seif et al. [58] in 2004 found similar changes to urodynamic parameters. Herzog et al. [59,60] in 2006 and 2008 reported similar results while showing through PET studies that STN DBS may be achieving these effects by improving cortical control over the micturition pathway. Pietraszko et al. [61] demonstrated that in addition to the quantitative change in volumes, these patients can experience significant qualitative improvements to urgency, frequency, nocturia, and hesitancy as well. In the case series by Halim et al. [51] discussed previously, the same EOPD patients who experienced improvements in hyperhidrosis also reported subjective improvements in bladder function. As with hyperhidrosis, the LOPD patients did not experience urinary improvement either. Fritsche et al. [62] reported two cases of acute urinary retention as a complication of STN DBS in patients who did not have LUTS prior to surgery. Winge et al. [63] in 2012 reported that STN DBS is at least comparable to medication in relieving LUTS, and superior in relieving nocturia.

Other targets for modulating micturition and urodynamic parameters include the GPi, the PAG, the VPL, and the VIM [64,65,66]. While GPi DBS can also ameliorate detrusor overactivity in patients with dystonia, it has been shown to worsen maximum flow rate and post-void residual volume [64]. In cystometric experiments on NP patients in which bladders are filled with isotonic saline via catheter infusion, Green et al. showed that PAG DBS dramatically increases the maximum cystometric capacity (MCC), the volume at which the patients would ask for the saline infusion to be stopped, but does not affect the volumes at which voiding is desired: V_ID_, strong desire (V_SD_), very strong desire (V_VSD_). By controlling for bladder sensation and pain, which were unchanged between stimulation on and off, they showed that the mechanism is most likely due to interruption of micturition directly [65]. In the same study, there were two VPL DBS patients who experienced smaller MCC volumes with stimulation on [65]. This is in agreement with a prior 2008 study by Kessler et al. [66] in which it was shown that VIM DBS results in reduced volumes for V_ID_, V_SD_, and MCC. These results suggest an induction of the micturition pathway by thalamic DBS.

### 4.2. DBS and Gastrointestinal Dysfunction

There has also been some evidence for DBS-mediated improvement of gastrointestinal dysmotility, which is a common symptom in PD [15,67]. In a study of PD patients with STN DBS by Ciucci et al. [68] in 2008, it was shown that STN DBS can improve the pharyngeal stage of deglutition, with faster pharyngeal transit times and degrees of bolus clearance, but does not improve the oral stage of deglutition. This resulted in less aspiration during swallowing with stimulation on versus off, perhaps due to greater coordination of the swallowing process [68]. Silbergleit et al. [69] suggested that STN DBS improves the patients’ perception of improved swallowing, in addition to improved motor control during the swallowing of solid foods. In a randomized cross-over study of sixteen PD patients with bilateral STN DBS either on or off at random, Derrey et al. [70] determined that STN DBS can improve bolus transport in the esophagus by causing amplified peristalsis of the distal esophagus and improved relaxation of the lower esophageal sphincter. They suggested that this is mediated by a cholinergic effect. Using ^13^C-acetate breath testing in a study of 16 bilateral STN DBS PD patients, Arai et al. [71] demonstrated improved gastric emptying with STN stimulation on versus off. In a recent study of twenty PD patients, Krygowska-Wajs et al. [72] demonstrated that STN DBS can improve gastrointestinal motility. They observed frequency reductions from 50% to 25% for dysphagia, 35% to 15% for sialorrhea, 95% to 75% for constipation, and 85% to 50% for difficulties with defecation. The patients in the study by Pietraszko et al. [61] also reported significant improvements in the same parameters as well as abdominal pain and rectal burning during or after defecation. One of the EOPD patients from the study by Halim et al. [51] also reported marked improvement in her bowel function, consistent with the findings above.

## 5. Conclusions

In addition to its current status as the primary surgical treatment for movement disorders, DBS has emerging potential for use as a surgical therapy for various dysautonomias. At the very least, the autonomic effects of DBS mandate careful assessment of autonomic dysfunction in patients requiring the treatment in order to choose the appropriate target—when a choice is available—to avoid undesirable effects that may lead to significant morbidity. Beyond that careful target selection, DBS offers the opportunity for novel therapy modalities that are not possible with conventional medical therapy. Autonomic drugs, while some are “selective,” generally target receptors throughout the whole body when only a specific organ or tissue type is desired, such as the heart or the bladder. DBS has the potential to add more precision to the arsenal available to physicians. Patients with orthostatic hypotension, for example, often are normotensive when supine. Pharmaceutical treatment includes α/β-agonists and adrenergic prodrugs such as droxidopa, which all can cause supine hypertension in these patients. DBS in concert with an accelerometer or mercury switch activator (that can detect when the patient is upright) can raise the patient’s blood pressure only while standing, deactivating the stimulation while the patient is supine [28]. In patients with urinary retention or urinary incontinence, self-control over activation/deactivation of the pulse generator could allow the patients to turn the stimulation on or off depending on whether inhibition or induction of micturition is desired given their condition and where their electrodes have been placed. The technology of DBS is continuing to evolve, and adaptive DBS (aDBS) will soon be able to adjust stimulation intensity based on a patient’s real-time clinical condition [73,74,75]. In addition, there are several ongoing government- and privately-funded projects aimed at enhancing the specificity of brain electrical stimulation (e.g., DARPA ElectRx, GSK electroceuticals).

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
