# Peer review of "Effects of Deep Brain Stimulation on Autonomic Function"

_brainsci, 2016, doi:10.3390/brainsci6030033_

Round 1
Reviewer 1 Report
The authors presents a review paper on autonomic system “side” effects of DBS to therapeutic targets. The authors are matter of fact in the review and mainly summarize the clinical research ongoing in the field, as opposed to the nonclinical research. The article reads a bit rushed (result after result without interpretation), but generally provides a fair and balanced representation of the evidence. The review tends to focus on solely clinical results and does not dig a bit deeper for the related nonclinical research to pair. Not sure what the editor would like to see, but a reader would like to not just see a summary of clinical results, but also where they fit in the research of, in particular, the brain regions discussed in all sections (it is discussed in DBS for respiratory function).
Additionally, in the discussion and future directions the authors make the point that drugs are non-specific, and I would agree with this assessment. However, this review is detailing how electrical stimulation for one disease has effects on others = non-specific. This hypocritical statement should be discussed a bit more, perhaps adding that there are many on-going government funded projects and privately funded projects to make electrical stimulation specific (DARPA ElectRx, GSK electroceuticals, etc. SPARC).
Methods,
Literature review criteria should be discussed in more detail...
Introduction
please place citations for specific diseases/disorder right after the disorder is mentioned - for clarity.
do not refer to the editors of the journal directly - this appears to incite a conflict of interest (I have no idea if there is, but certainly you do not want to present yourself as cozy with the editors)
"research available" - research is not available, the results are available - grammatical, but important
DBS and Cardiorespiratory Control
In general, this section seems to summarize another review, Hyam et al. with little (that I can tell) new interpretation. There is a statement in this section about higher frequencies spreading further in the thalamus to hypothalamus... The reference Hyam et al, mentions higher frequency AND higher total energy delivery. Biophysically, it is very unclear how higher frequency would lead to a larger electric field, but a higher voltage would, which is most of the time what the clinician adjusts. Please refer to higher total energy instead of higher frequency.
DBS and Sudomotor Control
No comments
DBS and Micturition
This section is particularly helpful in comparison / contrast with the Hyam 2012 paper.
Author Response
Reviewer 1 comments
The authors presents a review paper on autonomic system “side” effects of DBS to therapeutic targets. The authors are matter of fact in the review and mainly summarize the clinical research ongoing in the field, as opposed to the nonclinical research. The article reads a bit rushed (result after result without interpretation), but generally provides a fair and balanced representation of the evidence. The review tends to focus on solely clinical results and does not dig a bit deeper for the related nonclinical research to pair. Not sure what the editor would like to see, but a reader would like to not just see a summary of clinical results, but also where they fit in the research of, in particular, the brain regions discussed in all sections (it is discussed in DBS for respiratory function).
Additionally, in the discussion and future directions the authors make the point that drugs are non-specific, and I would agree with this assessment. However, this review is detailing how electrical stimulation for one disease has effects on others = non-specific. This hypocritical statement should be discussed a bit more, perhaps adding that there are many on-going government funded projects and privately funded projects to make electrical stimulation specific (DARPA ElectRx, GSK electroceuticals, etc. SPARC).
Response: Thank you for these comments. In the Future Directions section, we have added the statement that “There are several ongoing government- and privately-funded projects aimed at enhancing the specificity of brain electrical stimulation therapy (e.g. DARPA ElectRx, GSK electroceuticals).”
Methods,
Literature review criteria should be discussed in more detail...
Response: We have clarified our methodology as follows (1st page):
Methodology: A PubMed search of the available literature describing the autonomic effects of DBS was conducted through EndNote using the keywords listed above, along with any applicable iterations, in order to capture as many relevant references as possible. 99 references were found with this method. Those references were then categorized by the affected autonomic function and then further by DBS target.
Introduction
please place citations for specific diseases/disorder right after the disorder is mentioned - for clarity.
Response: Done.
do not refer to the editors of the journal directly - this appears to incite a conflict of interest (I have no idea if there is, but certainly you do not want to present yourself as cozy with the editors)
Response: We have removed mention of the editors in the text and acknowledgements.
"research available" - research is not available, the results are available - grammatical, but important
Response: We altered this Abstract sentence accordingly to “relevant available studies” thanks.
DBS and Cardiorespiratory Control
In general, this section seems to summarize another review, Hyam et al. with little (that I can tell) new interpretation. There is a statement in this section about higher frequencies spreading further in the thalamus to hypothalamus... The reference Hyam et al, mentions higher frequency AND higher total energy delivery. Biophysically, it is very unclear how higher frequency would lead to a larger electric field, but a higher voltage would, which is most of the time what the clinician adjusts. Please refer to higher total energy instead of higher frequency.
Response: Thank you for this comment. We agree and have clarified this by adding “and higher total energy delivery” and “and total energy delivery” in the appropriate sentences of this section (in Track Changes mode).
DBS and Sudomotor Control
No comments
DBS and Micturition
This section is particularly helpful in comparison / contrast with the Hyam 2012 paper.
Response: We thank the reviewer for this comment.
Reviewer 2 Report
This is a very interesting and well written review describing the effects of deep brain stimulation on autonomic regulation in a variety of patients and it presented information that is novel and clinically useful.
Author Response
Reviewer 2 comments
This is a very interesting and well written review describing the effects of deep brain stimulation on autonomic regulation in a variety of patients and it presented information that is novel and clinically useful.
Response: We thank the reviewer for this comment.